# Downregulation of OIP5-AS1 inhibits apoptosis in myocardial ischemia/reperfusion injury via modulating the MiR-145-5p/ROCK1 axis

**Jingyan Yang[1]◉, Jing Liu[2]◉, Xiaobo Liu[3], Dongling Xu[2], Juan Zhang◉[2]***

**1** Department of Pathology, The Second Hospital of Shandong University, Jinan, Shandong Province, China, **2** Department of Cardiology, The Second Hospital of Shandong University, Jinan, Shandong Province, China, **3** Shandong Blood Center, Jinan, Shandong Province, China

◉ These authors contributed equally to this work.
\* awwa6940@sina.com

## Abstract

### Purpose

The role of Long noncoding RNA OIP5-AS1 in myocardial ischemia/reperfusion (I/R) injury-induced apoptosis remains to be fully elucidated. The present study was conducted with the objective of investigating the function of OIP5-AS1 in myocardial I/R injury and exploring its potential mechanisms.

### Methods

In order to simulate the conditions of I/R, H9c2 cells were cultured in hypoxic/reoxygenated environments. Induction of I/R in Sprague-Dawley rats was achieved by ligating the left anterior descending coronary artery for 30 minutes followed by 180 minutes of reperfusion. OIP5-AS1 expression levels were assessed, and the degree of apoptosis was evaluated by TUNEL staining. Bioinformatic analysis was conducted to predict the interaction between microRNA-145-5p (miR-145-5p) and OIP5-AS1, and the expression levels of miR-145-5p and ROCK1 were determined.

### Results

Elevated levels of OIP5-AS1 were observed in H/R-treated H9c2 cells and in rat I/R models. Elevated OIP5-AS1 expression was associated with an increased incidence of apoptosis. The silencing of OIP5-AS1 in I/R conditions resulted in a significant suppression of cell apoptosis, reduced cleavage of caspase-3, decreased Bax levels, and increased Bcl-2 levels. Bioinformatic analysis predicted binding sites between miR-145-5p and OIP5-AS1. Furthermore, depletion of OIP5-AS1 in I/R conditions resulted in a substantial increase in miR-145-5p expression and a decrease in ROCK1 expression. The suppression of miR-145-5p reversed the effects of OIP5-AS1 depletion in I/R conditions.

**Data availability statement:** All data underlying the findings described in this manuscript are fully available without restriction in the Supporting information files accompanying this article.

**Funding:** Natural Science Foundation of Shandong Province (grant no. ZR2022MH285). The funders had no role in study design, data collection and analysis, decision to publish, or preparation of the manuscript.

**Competing interests:** The authors declare no conflicts of interest related to this study.

## Conclusions

Downregulation of OIP5-AS1 may prevent apoptosis in myocardial I/R injury by modulating the miR-145-5p/ROCK1 axis.

## Introduction

Worldwide, acute myocardial infarction (AMI) is a prevalent condition and is associated with significant mortality [1]. Primary percutaneous coronary intervention (PPCI), which involves the rapid restoration of blood flow to the ischemic myocardium, is the most effective treatment for AMI. However, the restoration of blood flow can paradoxically result in additional myocardial damage, known as myocardial ischemia-reperfusion (I/R) injury [2]. Myocardial I/R injury can lead to a range of complications, including arrhythmias, myocardial stunning, microvascular dysfunction, no-reflow phenomenon, and lethal reperfusion injury [3]. The pathophysiology of myocardial I/R injury involves rapid changes in pH, oxidative stress, mitochondrial dysfunction, inflammatory responses, calcium overload, metabolic alterations, and cardiomyocyte apoptosis [4–9]. Despite these insights, the molecular mechanisms underlying myocardial I/R injury remain complex and are not yet fully understood. Consequently, the development of therapies that target the molecular mechanisms of I/R is imperative to prevent I/R injury. A comprehensive elucidation of molecular networks and the development of multi-target therapeutic strategies may represent a critical strategy for overcoming the persistent challenges associated with I/R injury treatment.

Long non-coding RNAs (lncRNAs), which are longer than 200 nucleotides and lack protein-coding capabilities, have emerged as novel regulators and coordinators of gene expression in the past decade. This domain had previously been considered to be transcriptional noise [10–12]. It is becoming increasingly evident that lncRNAs may participate in an array of additional biological and pathological processes, including cardiovascular diseases [13]. The Opa-interacting protein-5 antisense transcript (OIP5-AS1) is evolutionarily conserved and is predominantly expressed in the cytoplasm. It has been proposed to regulate various physiological processes, such as mitosis, proliferation, and apoptosis [14].

Recent research indicates that a significant number of lncRNAs have been identified as important factors in several cardiac diseases, including myocardial I/R injury, primarily through the sequestration of microRNAs (miRNAs) [15,16]. Preliminary evidence suggests that OIP5-AS1 may be a candidate in the regulation of cardiomyocyte apoptosis in myocardial I/R injury [17,18]. As demonstrated in our previous study, overexpression of miR-145-5p, which targets Rho-associated coiled-coil-containing kinase 1 (ROCK1), has been shown to attenuate myocardial I/R-induced apoptosis [19]. Based on bioinformatics software (Starbase), OIP5-AS1 contains a putative binding site for miR-145-5p, suggesting that OIP5-AS1 may be involved in myocardial I/R injury via miR-145-5p. The objective of the present study was to investigate the role of OIP5-AS1 in myocardial I/R-induced apoptosis. The findings revealed that OIP5-AS1 sponged miR-145-5p to upregulate ROCK1 expression and accelerate apoptosis in myocardial I/R injury.

## Materials and methods

### Myocardial I/R injury cell model

The H9c2 myocardial cell line, obtained from the American Type Culture Collection (cat. no. CRL-1446), was cultured in Dulbecco's Modified Eagle Medium (DMEM; Hyclone; Cytiva) within a humidified incubator maintained at 37°C with an atmospheric composition of 95% oxygen ($O_2$) and 5% carbon dioxide ($CO_2$). In order to simulate myocardial I/R injury in vitro, hypoxia was induced by incubating the H9c2 cells in DMEM devoid of glucose and supplemented with 95% nitrogen ($N_2$) and 5% $CO_2$. Following a 6-hour hypoxic treatment, the cells were subjected to reoxygenation under normoxic conditions (with glucose; 5% $CO_2$) for an a further 6 hours. The control groups comprised of H9c2 cells that were cultured under standard conditions.

### Cell transfection

In order to achieve gene silencing of OIP5-AS1, small interfering RNAs (siRNAs) were deployed. SiRNAs targeting OIP5-AS1 (si-OIP5-AS1) and negative control siRNAs (si-NC) were designed and synthesized by RiboBio Company (China). MiR-145-5p mimics (miR10000851) and inhibitors (miR20000851) were also synthesized by RiboBio Company (China), with scrambled RNAs serving as negative controls for miR-145-5p mimics (mimic-NC, miR1N0000001-1-5) and miR-145-5p inhibitors (inhibitor-NC, miR2N0000001-1-5). The H9c2 cardiomyocytes were seeded in 6-well plates (5x10^5 cells/well) and transfected with si-OIP5-AS1(50 nM), miR-145-5p mimic (25 nM), miR-145-5p inhibitor (50 nM) or the respective controls at 37°C for 24 h. Following this, reverse transcription-quantitative polymerase chain reaction (RT-qPCR) was performed to select the most efficient mimic and inhibitor sequences. Optimization studies were performed using a cytotoxic-inducing siRNA (a non-targeting siRNA with a known cytotoxic effect) to establish the optimal transfection conditions while minimizing cell toxicity. The final transfection conditions were selected based on achieving a knockdown efficiency exceeding 80% while maintaining cell viability above 90%. Lipofectamine 2000 (Invitrogen, USA) was the reagent of choice for cell transfection. The ROCK1 sequence was sub-cloned into a pcDNA3.1 vector (ThermoFisher, USA) to generate a ROCK1-expression vector (1 µg/µl) and transfected into the H9c2 cardiomyocytes using Lipofectamine® 2000 at 37°C for 48 h. Cells transfection with the empty pcDNA3.1 vector were used as the negative control. Sequences of transfection oligonucleotides are presented in Table 1.

### Extraction of total RNA and RT-qPCR

Total RNA was extracted from cultured cells using TRIzol reagent (TianGen, China). RNA concentration and purity were quantified using a NanoDrop 2000 spectrophotometer (ThermoFisher, USA). The reverse transcription process was performed with a FastKing cDNA Synthesis Kit (TianGen, China). Subsequent qPCR was carried out utilizing the Fast Start Universal SYBR Green I Kit (Roche, Switzerland). The fold changes were calculated employing the 2-ΔΔCq method, with Actin and U6 serving as internal reference controls. Bio-repeats were conducted in six replicates. The thermocycling conditions were as follows: initial denaturation at 95°C for 15 minutes followed by 40 cycles at 95°C for 5 seconds and 60°C for 30 seconds. The primer sequences used for RT-qPCR are detailed in Table 2.

**Table 1. Transfection sequences.**

| Oligo name | Sequence (5′→3′) | Final transfection concentration |
| --- | --- | --- |
| si-OIP5-AS1 | GGTTAGTCAGATTGGACAA | 50 nM |
| si-NC | TACCGACTGGCAATTCATG | 50 nM |
| miR-145-5p mimic | GUCCAGUUUUCCCAGGAAUCCCU | 25 nM |
| miR-145-5p inhibitor | AGGGAUUCCUGGGAAAACUGGAC | 50 nM |
| miR mimic-NC | UUCUCCGAACGUGUCACGUTT | 25 nM |
| miR inhibitor-NC | CAGUACUUUUGUGUAGUACAA | 50 nM |

**Table 2. Primers used in reverse transcription-polymerase chain reaction.**

| Primer name | Forward primer | Reverse primer |
| --- | --- | --- |
| OIP5-AS1 | AGGTGCAAGCATACCGTCTC | TCAACACAGCCCTCTGCATT |
| miR-145-5p | GGGGTCCAGTTTTCCCAG | AACTGGTGTCGTGGAGTCGGC |
| ROCK1 | GGAAACGCTCCGAGACACTG | CTGTTCTCACTGGGATTGCTG |
| Actin | TGCTATGTTGCCCTAGACTTCG | GTTGGCATAGAGGTCTTTACGG |
| U6 | CTCGCTTCGGCAGCACAT | AAATATGGAACGCTTCACG |

## Luciferase reporter assay

TargetScan and Starbase were instrumental in predicting potential targets of OIP5-AS1 and miRNA-145-5p. In order to confirm the direct binding interactions between OIP5-AS1/miRNA-145-5p and miRNA-145-5p/ROCK1, a dual-luciferase reporter gene analysis was conducted. Wild-type (WT) and mutant (MUT) OIP5-AS1-WT/MUT or ROCK1-WT/MUT luciferase plasmids were introduced into the pmirGLO dual luciferase miRNA-targeting vector (Promega, USA). These plasmids were cotransfected with miR-145-5p mimic or mimic NC. Luciferase activity was measured using the Dual-Luciferase System (Promega, USA).

## TUNEL staining

The One-step TUNEL FITC Apoptosis Detection Kit (APExBio, USA) was utilised in accordance with the manufacturer's protocol to detect cellular apoptosis. Fluorescence was observed under a microscope (Olympus, Japan) after counter-staining with 4,6-diamino-2-phenylindole (DAPI).

## Western blot

Protein samples were resolved by 10% sodium dodecyl sulfate–polyacrylamide gel electrophoresis (SDS-PAGE). Subsequently, the samples were transferred to polyvinylidene fluoride (PVDF) membranes and blocked. The PVDF membranes were then subjected to an incubation with primary antibodies, including mouse anti-β-Actin (66009–1, Proteintech, 1:20000), rabbit anti-cleaved-Caspase3 (25128–1, Proteintech, 1:1000), rabbit anti-Bcl-2 (12789–1, Proteintech, 1:1000), rabbit anti-Bax (50599–2, Proteintech, 1:1000), and rabbit anti-ROCK1 (21850–1, Proteintech, 1:5000) at 4°C overnight. Membranes were then incubated with horseradish peroxidase-conjugated secondary antibodies (goat anti-rabbit, ZB-2301, ZSGB-BIO, 1:2000; goat anti-mouse, ZB-2305, ZSGB-BIO, 1:2000) at 25°C for 1 hour. The relative expression levels were determined using a Gel-Pro analyser (Media Cybernetics).

## Myocardial I/R injury rat model

Male Sprague-Dawley rats (250-300g) were procured from Shandong University. Rats were anaesthetized with pentobarbital (50 mg/kg, intraperitoneal) and mechanically ventilated. A thoracotomy and pericardiotomy were performed, and the left anterior descending coronary artery (LAD) was occluded for 30 minutes to induce ischemia, followed by a 180-minute reperfusion period. Sham control animals underwent the same surgical procedure without LAD ligation. Following the conclusion of the experiments, the animals were euthanized with an overdose of isoflurane administered for approximately 10 minutes, followed by euthanasia through exsanguination. Efforts were made to minimize animal suffering and to use the minimum number of animals required to achieve statistical significance. All experiments involving animals were conducted in accordance with the Ethics Committee of Second Hospital of Shandong University for the care and use of laboratory animals, and all procedures were conducted in accordance with the Institutional Animal Care and Use Committee and National Institutes of Health guidelines. The present study was approved by the Institutional Animal Care and Use Committee of the Second Hospital of Shandong University (approval no. KYLL-2021(KJ)A-0503).

### Animal gene therapy

Adeno-associated virus 9 (AAV9) vectors containing short hairpin RNA (shRNA) targeting OIP5-AS1 (HBAAV9-CTNT-shOIP5-AS1) and the corresponding negative control (HBAAV9-CTNT-sh-NC) were constructed by Hanbio. Rats were subjected to gene transfer via the intravenous tail vein injection of 5×10^10 AAV9 particles. The knockdown efficiency of myocardial OIP5-AS1 was verified by RT-qPCR.

### Histological analysis

Tissues from the left ventricle (LV) were embedded in paraffin and sectioned at a thickness of 5 µm. TUNEL staining was used to assess the presence of apoptosis in cardiac tissue, with TUNEL-positive cells quantified as a percentage of total myocytes in ten randomly selected high-power fields per section.

### Statistical analysis

All descriptive variables are expressed as mean±SD. Normality was assessed using the Shapiro-Wilk test. Comparisons between two groups were conducted using the unpaired Student's t-test for normally distributed variables and the Mann-Whitney U test for non-normally distributed variables. One-way ANOVA followed by Tukey's post hoc test was used to analyze differences among multiple groups. Statistical analyses were performed using SPSS 20.0 (IBM Corp). A statistically significant difference was defined as $P < 0.05$.

## Results

### I/R induced apoptosis and OIP5-AS1 over-expression in cardiomyocytes

It is widely recognized that increased apoptosis in cardiomyocytes follows ischemia/reperfusion (I/R) injury. As illustrated in Fig 1A, the growth state of cardiomyocytes was observed under both normal conditions and I/R conditions. In the present experiment, the proportion of apoptotic cells was determined using TUNEL staining (Fig 1B), with untreated H9c2 cardiomyocytes serving as the control group. A significant increase in apoptosis was observed in the I/R group in comparison to the control group (Fig 1C). The expression level of OIP5-AS1 was examined by RT-qPCR, revealing that OIP5-AS1 was overexpressed in the I/R group compared to the control group (Fig 1D). Furthermore, the expression levels of miR-145-5p and ROCK1 were also assessed by RT-qPCR. MiR-145-5p expression was found to be significantly downregulated (Fig 1E), whereas ROCK1 expression was significantly upregulated in I/R compared with the control group (Fig 1F).

### Downregulation of OIP5-AS1 decreased I/R-induced cardiomyocytes apoptosis

In order to investigate the potential role of OIP5-AS1 in myocardial I/R, OIP5-AS1 was silenced in H9c2 cardiomyocytes using siRNAs (Fig 2A). The silencing OIP5-AS1 was found to result in the suppression of apoptosis, as identified by the monitoring of TUNEL staining (Fig 2B, C). Furthermore, the assessment of cell apoptosis was conducted by measuring the activity of Caspase-3, Bcl-2, and Bax. Caspase-3 promotes apoptosis, whereas Bcl-2 suppresses apoptosis [20]. Western blot assays showed that I/R upregulated the expression of cleaved-Caspase3 and Bax and downregulated Bcl-2 expression (Fig 2D). Following the silencing OIP5-AS1 in I/R, Caspase-3 cleavage and Bax were reduced, and Bcl-2 was upregulated. Collectively, the data showed that OIP5-AS1 was upregulated in I/R and silencing OIP5-AS1 resulted in a reduction in apoptosis.

### OIP5-AS1 bonded to miR-145-5p

LncRNA target gene predictions were conducted using StarBase (http://starbase.sysu.edu.cn/index.). The analysis predicted that OIP5-AS1 binds with miR-145-5p (Fig 3A). Luciferase reporter vectors containing either the wild-type

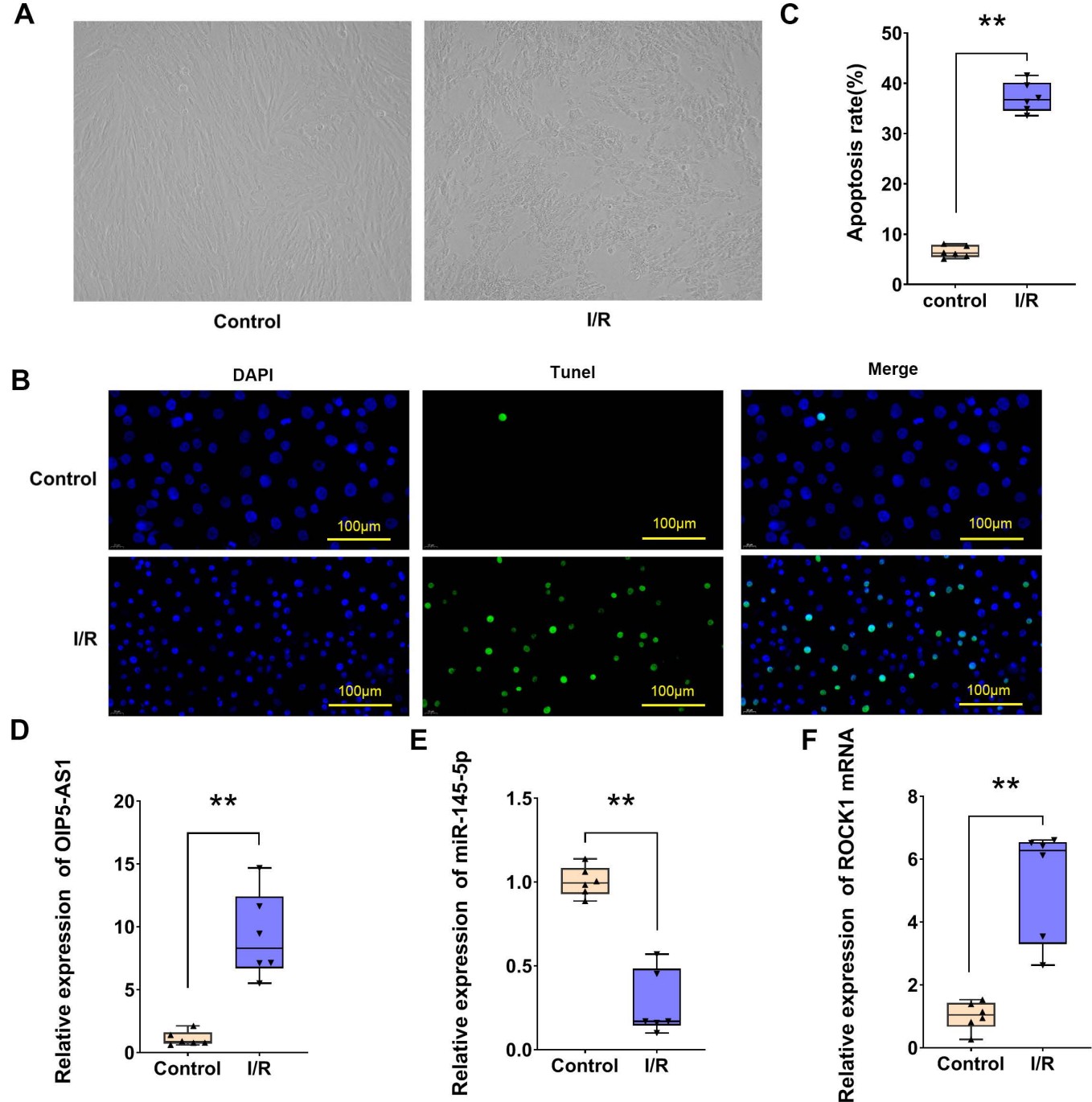

**Fig 1. Myocardial I/R injury induces apoptosis and upregulation of OIP5-AS1 in cardiomyocytes.** (A) H9c2 cardiomyocytes cultured under normoxic and hypoxia/reoxygenation conditions. (B) TUNEL staining for apoptosis in control and I/R groups. (C) Statistical representation of apoptosis rates. (D) RT-qPCR analysis of OIP5-AS1 expression in H9c2 cells. (E) RT-qPCR analysis of miR-145-5p expression in H9c2 cells. (F) RT-qPCR analysis of ROCK1 expression in H9c2 cells. **$P < 0.01$.

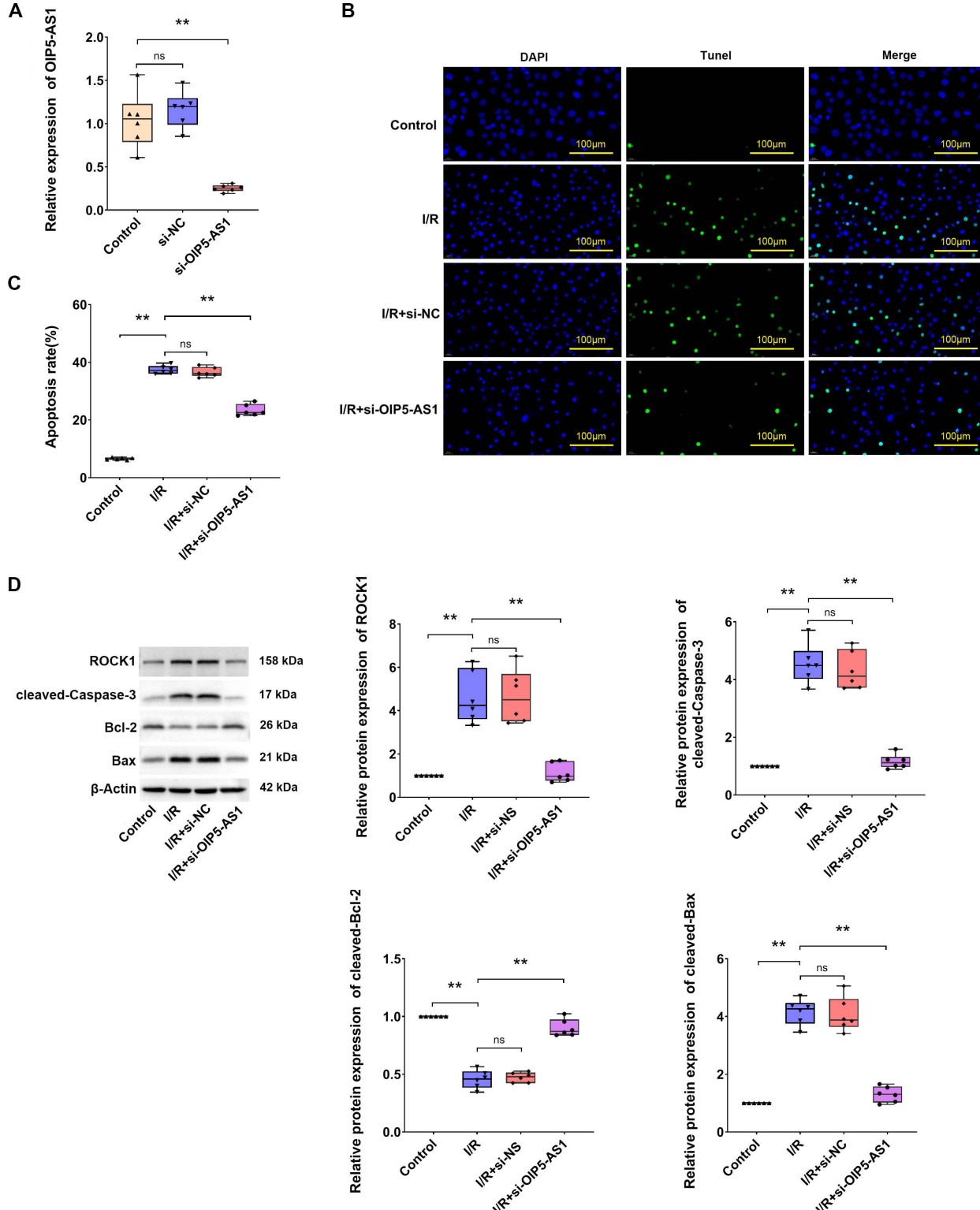

**Fig 2. Knockdown of OIP5-AS1 leads to reduced cell apoptosis in I/R cell models.** (A) Efficiency of si-OIP5-AS1 determined by RT-qPCR in H9c2 cells. (B) Representative TUNEL staining showing apoptosis of H9c2 cells. (C) Statistical representation of apoptosis rates. (D) Western blot analysis showing protein expressions of ROCK1, cleaved-Caspase3, Bcl-2, and Bax. **P < 0.01.

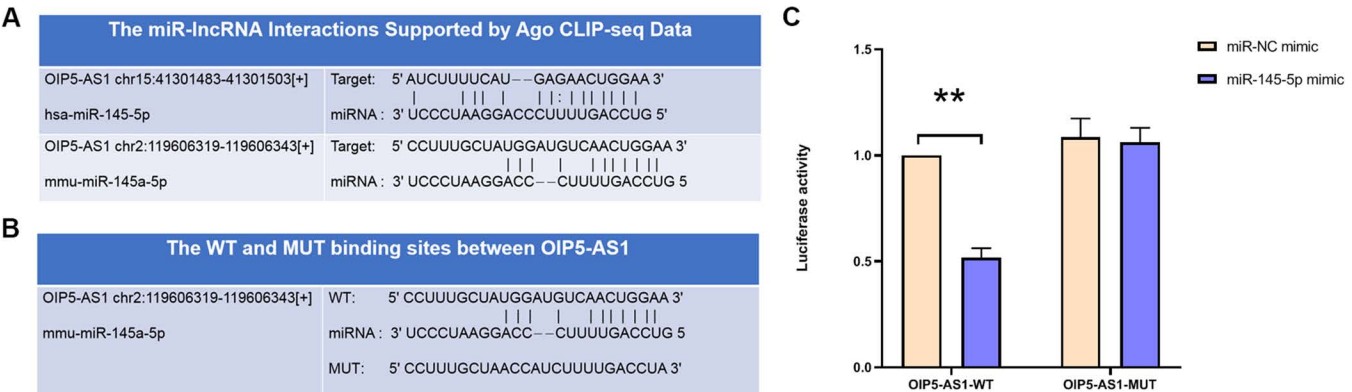

**Fig 3. OIP5-AS1 binds to miR-145-5p.** (A) StarBase prediction of binding sites between OIP5-AS1 and miR-145-5p. (B) WT and MUT binding sites between OIP5-AS1 and miR-145-5p. (C) Luciferase reporter assay verifying the interaction between OIP5-AS1 and miR-145-5p. **P<0.01.

OIP5-AS1 (OIP5-AS1-WT) or its mutant version with disrupted miR-145-5p binding sites (OIP5-AS1-MUT) were constructed to experimentally validate their interaction (Fig 3B). The luciferase activity of OIP5-AS1-WT was significantly reduced upon overexpression of miR-145-5p. Conversely, there were no changes in the luciferase activity of OIP5-AS1-MUT in response to the elevation of miR-145-5p (Fig 3C). These findings demonstrated the specific binding of OIP5-AS1 with miR-145-5p.

### MiR-145-5p directly targeted ROCK1

Bioinformatic analysis using TargetScan (http://www.targetscan.org) predicted ROCK1 as a high-confidence target of miR-145-5p (Fig 4A). In order to functionally validate ROCK1 as a direct target of miR-145-5p, dual-luciferase reporter assays were performed using the wild-type ROCK1 (ROCK1-WT) and the mutant constructs (ROCK1-MUT) (Fig 4B). MiR-145-5p mimic significantly reduced luciferase activity in ROCK1-WT-transfected cells but showed no effect on ROCK1-MUT activity (Fig 4C). These findings indicated that miR-145-5p specifically targeted ROCK1.

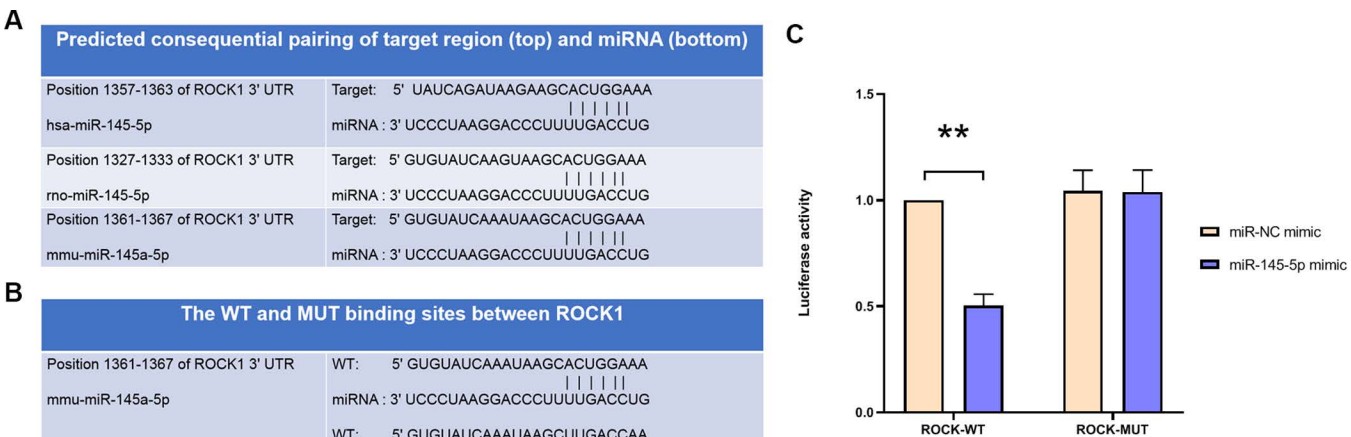

**Fig 4. ROCK1 is a target gene of miR-145-5p.** (A) TargetScan prediction of binding sites between ROCK1 and miR-145-5p. (B) WT and MUT binding sites between miR-145-5p and ROCK1. (C) Luciferase reporter assay verifying the interaction between ROCK1 and miR-145-5p. **P<0.01.

## OIP5-AS1 regulated cell apoptosis in myocardial I/R cell models via miR-145-5p

Compared with the control group, expression of miR-145-5p in H9c2 cardiomyocytes was significantly downregulated in I/R. Silencing of OIP5-AS1 in I/R significantly upregulated miR-145-5p expression (Fig 5A, B). As demonstrated collectively in Figs 3C and 5B, I/R-mediated miR-145-5p downregulation and its capacity to bind OIP5-AS1 were revealed. Furthermore, the present study investigated whether OIP5-AS1 regulated apoptosis induced by I/R via miR-145-5p. TUNEL staining revealed that repression of OIP5-AS1 significantly reduced cell apoptosis, and treatment with an inhibitor of miR-145-5p completely abolished the rescue of apoptotic cells (Fig 6A, B). Caspase-3 cleavage and Bax protein levels were decreased, and Bcl-2 expression was increased in I/R under OIP5-AS1 suppression. This effect was reversed by co-treatment of si-OIP5-AS1 and miR-145-5p inhibitor (Fig 6C). Collectively, the inhibition of apoptosis in I/R resulting from OIP5-AS1 reduction was reversed by silencing miR-145-5p. The results suggested that OIP5-AS1 modulated I/R-mediated apoptosis via miR-145-5p.

## ROCK1 overexpression reversed anti-apoptotic effects of OIP5-AS1 downregulation

In I/R injury, ROCK1 expression was significantly upregulated (Fig 1F). Conversely, suppression of OIP5-AS1 induced a marked reduction in ROCK1 protein and mRNA expression levels (Figs 2D and 5C). To further elucidate the molecular mechanism by which OIP5-AS1 modulates cardiomyocyte apoptosis in myocardial I/R injury, we performed gain-of-function experiments using ROCK1 overexpression in OIP5-AS1-deficient cells. Quantitative analysis revealed that ROCK1 overexpression significantly attenuated the anti-apoptotic effect of OIP5-AS1 downregulation (Fig 7A, B). The results suggested that ROCK1 played a significant role in mediating OIP5-AS1's effects on cardiomyocyte apoptosis in I/R.

## OIP5-AS1 regulated apoptosis in myocardial I/R cell models by targeting miR-145-5p/ROCK1 axis

It has been demonstrated that miR-145-5p overexpression attenuated H9c2 cardiomyocyte apoptosis induced by I/R through the targeting of ROCK1 [19]. In the present study, we initially corroborated the efficacy of OIP5-AS1 knockdown in our experimental models (Figs 5A and 8A). Importantly, OIP5-AS1 deficiency led to a substantial decrease in ROCK1 expression, accompanied by an increase in miR-145-5p levels under I/R conditions (Figs 5B, C and 8B, C). Furthermore, the reduced expression of ROCK1, consequent to the suppression of OIP5-AS1, was ound to be reversible following the inhibition of miR-145-5p (Fig 8C). Competing endogenous RNAs (ceRNAs) are a class of regulatory transcripts that modulate gene expression through microRNA competition, whereby they sequester shared microRNAs and prevent their

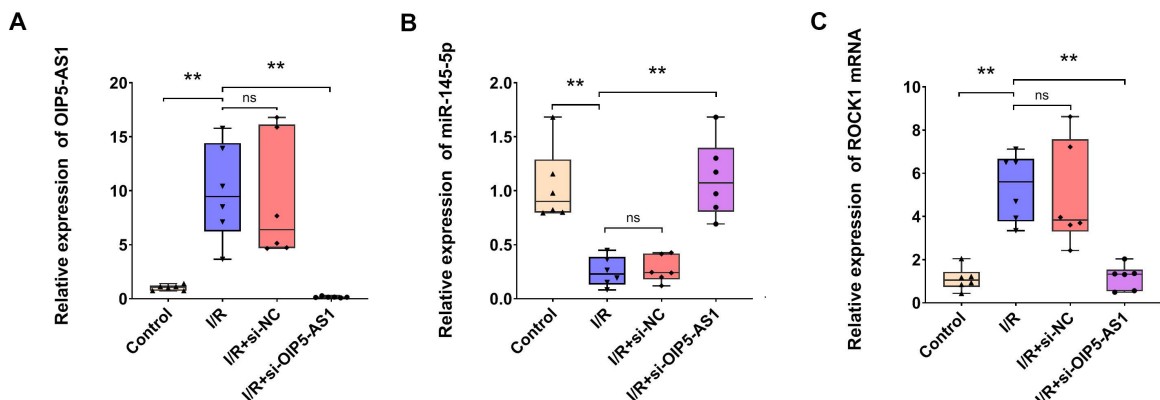

**Fig 5. OIP5-AS1 silencing rescues miR-145-5p and suppresses ROCK1.** (A) RT-qPCR analysis of OIP5-AS1 expression in H9c2 cells. (B) RT-qPCR analysis of miR-145-5p expression in H9c2 cells. (C) RT-qPCR analysis of ROCK1 expression in H9c2 cells. **P<0.01.

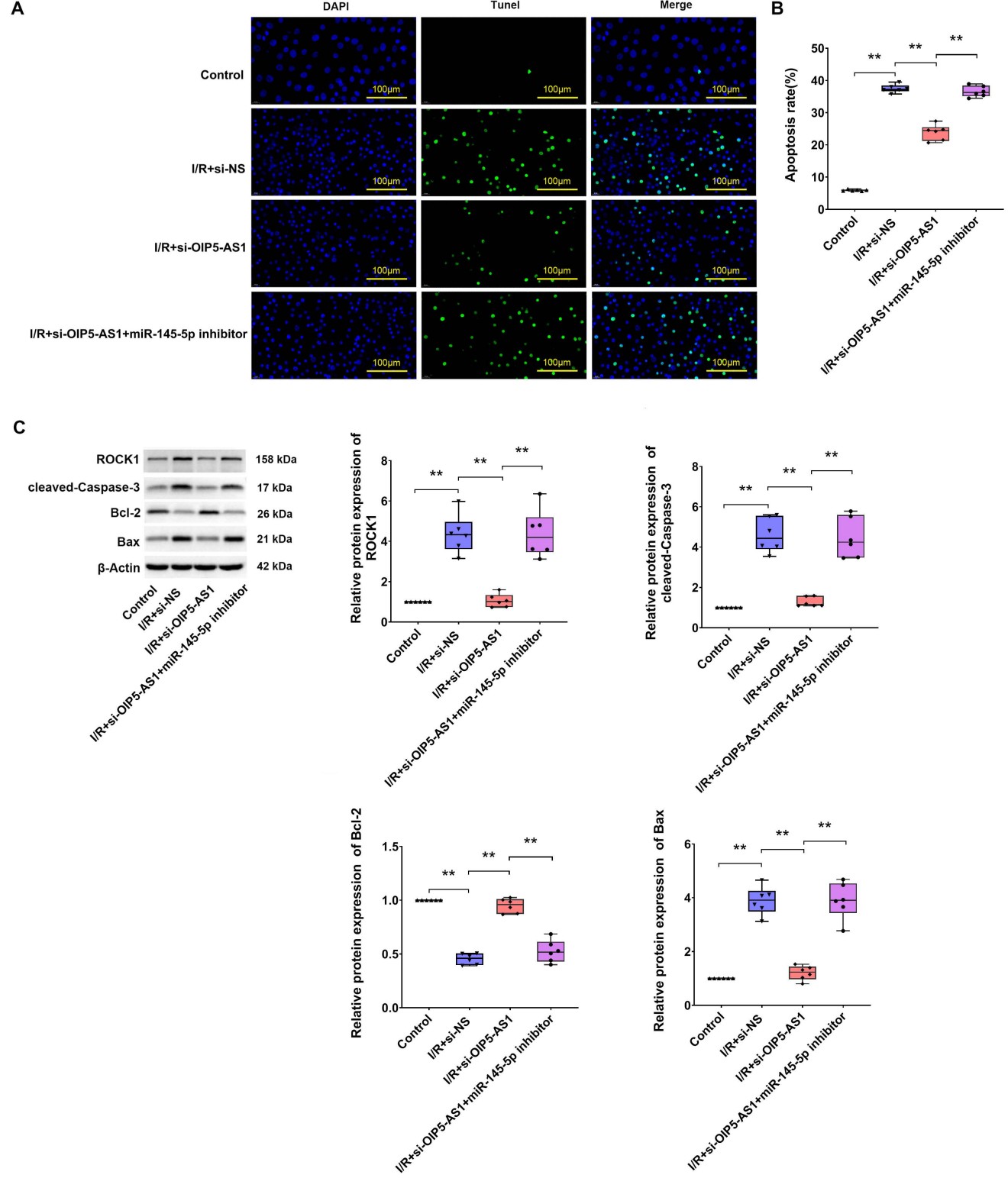

**Fig 6. OIP5-AS1 modulates cell apoptosis via miR-145-5p.** (A) Representative TUNEL staining showing apoptosis of H9c2 I/R cells. (B) Statistical representation of apoptosis rates. (C) Western blot analysis showing protein expressions of ROCK1, cleaved-Caspase3, Bcl-2, and Bax. **P<0.01.

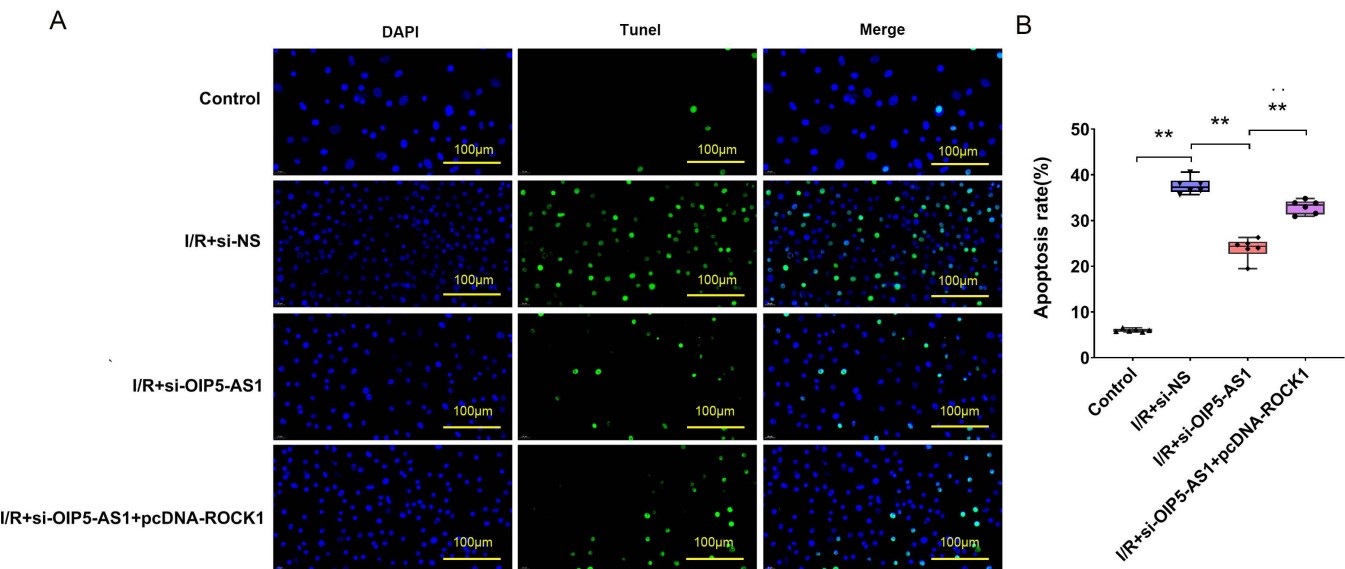

**Fig 7. ROCK1 reverses OIP5-AS1 silencing-mediated apoptosis suppression.** (A) Representative TUNEL staining showing apoptosis. (B) Statistical representation of apoptosis rates. **P<0.01.

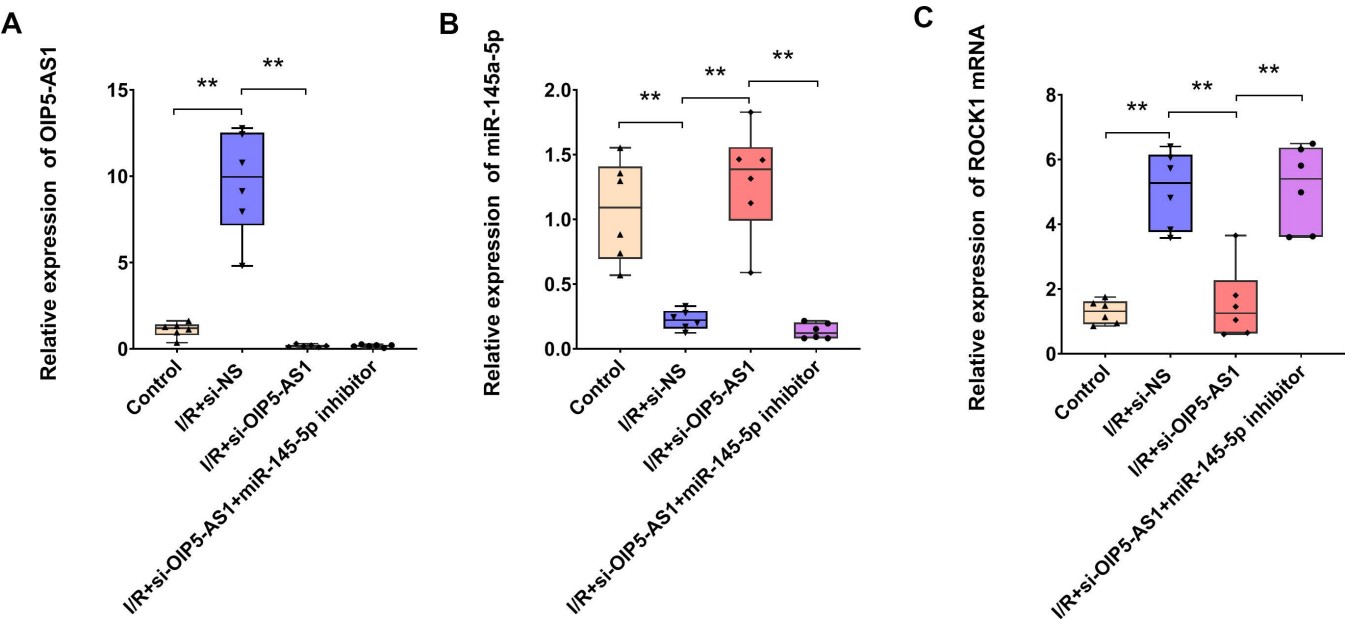

**Fig 8. MiR-145-5p inhibition rescues ROCK1 expression following OIP5-AS1 silencing in I/R Injury.** (A) RT-qPCR analysis of OIP5-AS1 expression in H9c2 cells. (B) RT-qPCR analysis of miR-145-5p expression in H9c2 cells. (C) RT-qPCR analysis of ROCK1 expression in H9c2 cells. **P<0.01.

binding to target mRNAs [21]. Collectively, these data indicated that OIP5-AS1 may function as a ceRNA against miR-145-5p, thereby promoting ROCK1 upregulation. OIP5-AS1 was found to modulate cell apoptosis in I/R cell models by acting on the miR-145-5p/ROCK1 axis.

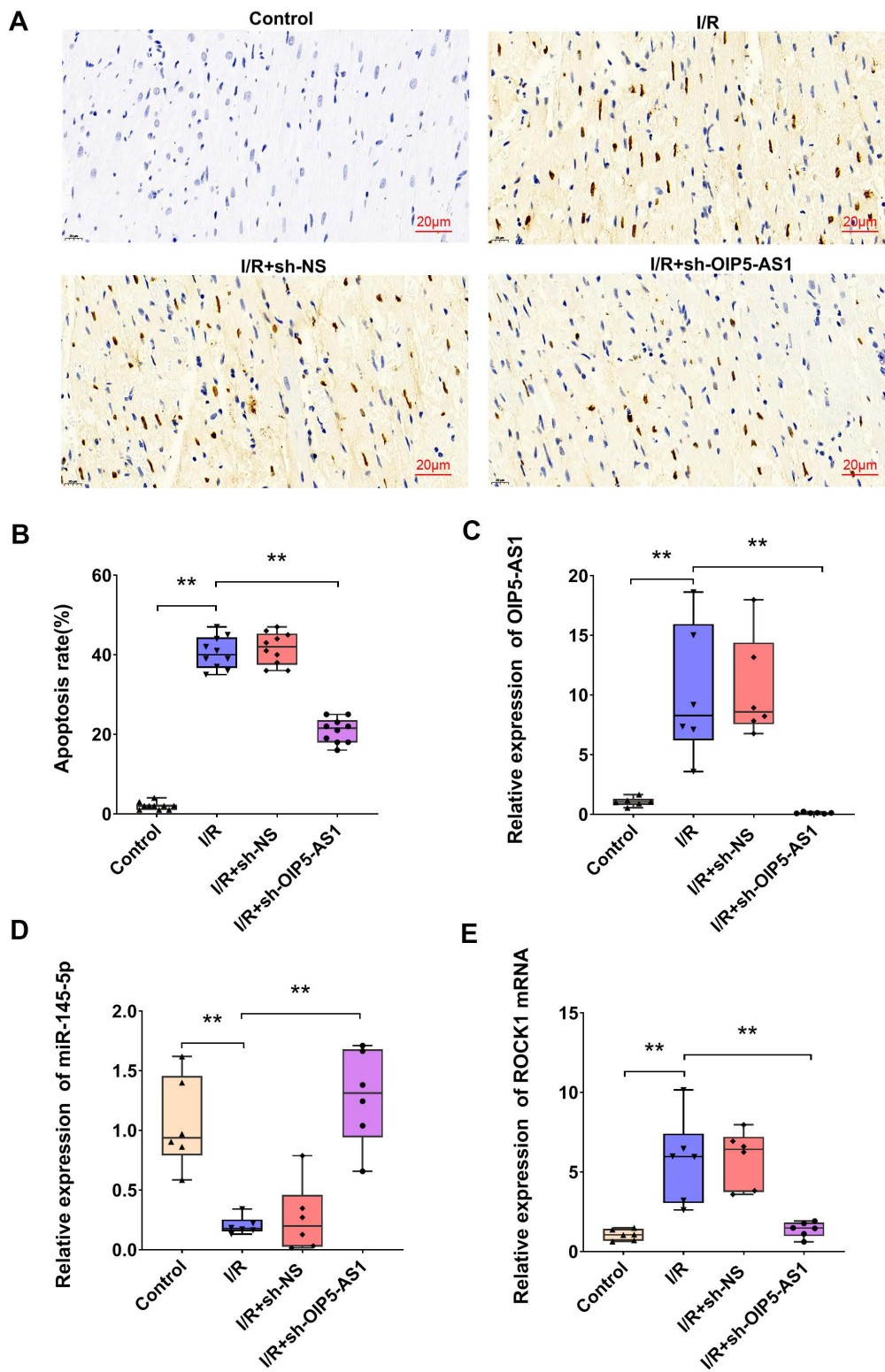

**Fig 9. Silencing of OIP5-AS1 attenuates apoptosis by restoring miR-145-5p and suppressing ROCK1 in I/R rat models.** (A) Representative TUNEL staining showing apoptosis. (B) Statistical representation of apoptosis rates. (C) RT-qPCR analysis of OIP5-AS1 expression in I/R rat models. (D) RT-qPCR analysis of miR-145-5p expression in I/R rat models. (E) RT-qPCR analysis of ROCK1 expression. **$P < 0.01$.

### Downregulation of OIP5-AS1 alleviated cardiomyocyte apoptosis in myocardial I/R rats

In order to investigate the role of OIP5-AS1 in I/R-induced cardiomyocyte apoptosis in vivo, myocardial I/R rat models were subjected to histopathological and molecular analyses. TUNEL staining revealed a pronounced increase in apoptotic nuclei (brown-stained) compared to healthy nuclei (blue-stained) in I/R-injured myocardium, with quantification demonstrating a significant elevation in the apoptotic index (Fig 9A, B). Notably, OIP5-AS1 knockdown significantly attenuated I/R-induced apoptosis confirming its pro-apoptotic function in this context. Molecular profiling of I/R myocardium revealed dysregulation of a putative OIP5-AS1-miR-145-5p-ROCK1 axis: OIP5-AS1 was significantly overexpressed, miR-145-5p expression was markedly downregulated, and ROCK1 expression was substantially upregulated (Fig 9C–E). OIP5-AS1 silencing in I/R rats resulted in a significant increase in miR-145-5p expression and a significant decrease in ROCK1 expression. These findings indicated that OIP5-AS1 may regulate apoptosis in I/R rat models via the miR-145-5p/ROCK1 axis.

## Discussion

Recent studies have highlighted the significant role of lncRNAs and miRNAs in myocardial I/R injury [22]. A number of studies have confirmed that various lncRNAs play a critical role in I/R-induced cardiomyocyte apoptosis [23–25]. However, the function of OIP5-AS1 in myocardial I/R injury remain to be elucidated. The present study therefore sought to investigate the impact of OIP5-AS1 on apoptosis during myocardial I/R injury. Niu et al. hypothesised that OIP5-AS1 could potentially mitigate myocardial I/R injury through the sponge effect on miR-29a [18]. In contrast to their findings, our research has confirmed that OIP5-AS1 is upregulated following myocardial I/R injury and exacerbates I/R injury. Furthermore, OIP5-AS1 was found to be upregulated in both H9c2 cells and rat models of myocardial I/R injury. The present study demonstrates that the knockdown of OIP5-AS1 significantly attenuates apoptosis during I/R injury in both in vivo and in vitro models. The results obtained in this study lend support to the hypothesis that OIP5-AS1 may play a critical role in the apoptosis of cells induced by I/R injury in the myocardium.

Numerous studies have investigated the regulation of cardiomyocyte apoptosis in I/R injury by lncRNAs through the sequestration of specific miRNAs [26–29]. Consequently, the present study hypothesized whether OIP5-AS1 could modulate miRNAs in myocardial I/R injury. By bioinformatics analysis and experimental validation, we have identified a binding site for miR-145-5p within the OIP5-AS1 sequence. Our findings indicate that OIP5-AS1 negatively regulates the cardioprotective effects of miR-145-5p through direct interaction, as demonstrated by our mechanistic studies. In our previous study, we reported that miR-145-5p was downregulated following myocardial I/R injury, and that overexpression of miR-145-5p protected H9c2 cells from apoptosis induced by I/R [19]. The expression of miR-145-5p was upregulated when OIP5-AS1 was silenced in cardiomyocytes using siRNA. Silencing of OIP5-AS1 could suppress cell apoptosis and modulate the expressions of apoptosis-associated factors, including caspase-3, Bax, and Bcl-2. Downregulating OIP5-AS1 may protect cardiomyocytes from apoptosis during I/R injury by upregulating miR-145-5p. It is noteworthy that the inhibitor of miR-145-5p significantly reduced the inhibition of cell apoptosis induced by the downregulation of OIP5-AS1 after I/R injury. This finding provides further evidence to support the competitive binding relationship between OIP5-AS1 and miR-145-5p, in which OIP5-AS1 acts as a molecular sponge for miR-145-5p.

ROCK, an essential effector of the RhoA GTPase, is comprised of two closely related isoforms: ROCK1 and ROCK2 [30]. ROCK has been implicated in the aberrant pathological mechanisms of cardiac diseases, including myocardial ischemia/reperfusion I/R injury [31,32]. ROCK1 and ROCK2 have been shown to exert distinct roles in various tissues and cell types [31,33]. In view of the finding that ROCK1 is targeted by miR-145-5p [19,34], it was concluded that OIP5-AS1 functions as a ceRNA for miR-145-5p to regulate ROCK1. The present study revealed that the knockdown of OIP5-AS1 reduced ROCK1 expression and apoptosis induced by I/R. However, this reduction was counteracted by decreased levels of miR-145-5p. The findings of this study indicate that miR-145-5p exhibits a protective effect against myocardial I/R injury, while OIP5-AS1 has a deleterious effect on I/R injury. Downregulation of OIP5-AS1 may impede apoptosis in myocardial I/R injury by modulating the miR-145-5p/ROCK1 axis.

Despite the findings of this study, which elucidate the role of the OIP5-AS1/miR-145-5p/ROCK1 axis, several limitations remain. Initially, the experimental phase was conducted in H9c2 cells, with preliminary validation occurring in a rat model. Nevertheless, further validation in more clinically relevant models is necessary. Secondly, the knockdown of OIP5-AS1 relied on siRNA or shRNA transfection, which may have off-target effects. Consequently, further validation employing multi-target approaches, such as CRISPR-Cas9 gene knockout, is imperative to substantiate specificity [35]. Furthermore, the interaction network of OIP5-AS1 with other miRNAs or proteins has not been comprehensively analyzed, potentially overlooking key regulatory pathways. In future studies, the objective is to validate the role of OIP5-AS1 in animal models and analyze its expression and prognostic relevance in clinical samples, such as plasma or myocardial biopsies from acute myocardial infarction patients, with a view to enhancing clinical translational potential.

## Conclusions

In summary, we have unveiled a novel mechanism by which OIP5-AS1, acting as a molecular sponge for miR-145-5p, modulates I/R-induced apoptosis via ROCK1, thereby providing a potential therapeutic target for cardioprotection. Future research endeavors should focus on further substantiating its translational potential and addressing the associated technological challenges.

## Supporting information

**S1 File. Experimental data.**
(XLSX)

**S2 File. Melting curves.**
(TIF)

**S3 File. Original blot and gel image data.**
(TIF)

## Author contributions

**Conceptualization:** Juan Zhang.

**Data curation:** Jingyan Yang, Jing Liu.

**Formal analysis:** Jingyan Yang, Jing Liu, Xiaobo Liu.

**Project administration:** Juan Zhang.

**Writing – original draft:** Juan Zhang, Dongling Xu.

**Writing – review & editing:** Juan Zhang.

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
