## [Decision Letter · Decision Letter 0]

19 Mar 2025

PONE-D-25-10451Downregulation of  OIP5-AS1 Inhibits Apoptosis in Myocardial Ischemia/Reperfusion Injury via Modulating the MiR-145-5p/ROCK1 AxisPLOS ONE

Dear Dr. Zhang,

Thank you for submitting your manuscript to PLOS ONE. After careful consideration, we feel that it has merit but does not fully meet PLOS ONE’s publication criteria as it currently stands. Therefore, we invite you to submit a revised version of the manuscript that addresses the points raised during the review process.

Please check the reviewers' comments. Ensure that all figures are appropriately cited and that the methods are described in sufficient detail for reproducibility. Include catalog IDs for all products used, as well as the sequences for the miR-145-5p mimic. Specify which method was used for the 2^-ΔΔCq calculation (Livak, Pfaffl, or another approach) and provide the corresponding citation. Clearly describe the procedure for adaptor ligation during miRNA-to-cDNA conversion. Define all abbreviations upon their first mention and thoroughly review the English grammar throughout the document. Include melting curves to confirm the specific amplification of all target regions, including miR-145-5p. Replace bar plots with box plots or violin plots, supplemented by point or jitter plots to display all biological replicates, particularly for expression analyses, as this provides a more transparent representation of the results.

We look forward to receiving your revised manuscript.

Kind regards,

Alexis G. Murillo Carrasco

Academic Editor

PLOS ONE

“Natural Science Foundation of Shandong Province (grant no. ZR2022MH285)”

4. Thank you for stating the following in the Funding Section of your manuscript:

“The authors gratefully acknowledge research support provided by Natural Science Foundation of Shandong Province (grant no. ZR2022MH285).”

“Natural Science Foundation of Shandong Province (grant no. ZR2022MH285)”

In your cover letter, please note whether your blot/gel image data are in Supporting Information or posted at a public data repository, provide the repository URL if relevant, and provide specific details as to which raw blot/gel images, if any, are not available. Email us at plosone@plos.org if you have any questions

7. Your ethics statement should only appear in the Methods section of your manuscript. If your ethics statement is written in any section besides the Methods, please delete it from any other section.

Reviewers' comments:

Reviewer's Responses to Questions

**Comments to the Author**

1. Is the manuscript technically sound, and do the data support the conclusions?

Reviewer #1: Yes

Reviewer #2: Partly

Reviewer #3: Partly

2. Has the statistical analysis been performed appropriately and rigorously? 

Reviewer #1: Yes

Reviewer #2: Yes

Reviewer #3: Yes

3. Have the authors made all data underlying the findings in their manuscript fully available?

Reviewer #1: Yes

Reviewer #2: Yes

Reviewer #3: No

4. Is the manuscript presented in an intelligible fashion and written in standard English?

Reviewer #1: Yes

Reviewer #2: Yes

Reviewer #3: No

5. Review Comments to the Author

Reviewer #1: Yang and his colleagues observed that the expression of the long non-coding RNA (lncRNA) OIP5-AS1 and apoptosis were increased in myocardial ischemia/reperfusion (I/R) injury. Based on their previous study, they further confirmed that the expression of miR-145-5p was increased, while its target, ROCK1, was decreased. They then demonstrated that reducing OIP5-AS1 expression led to a decrease in apoptosis and experimentally validated the relationship between OIP5-AS1 and miR-145-5p.

The paper presents a very straightforward mechanism, and the experimental evidence supporting each claim is convincing.

However, to strengthen the authors' argument, I suggest including additional experiments to clarify the relationship between ROCK1 and OIP5-AS1:

1. Does overexpression of ROCK1 in the context of reduced OIP5-AS1 modulate the level of apoptosis?

2. How does apoptosis change when anti-miR-145-5p is introduced in a setting where OIP5-AS1 is downregulated?

Reviewer #2: Thank you for the opportunity to review your manuscript titled "Downregulation of OIP5-AS1 Inhibits Apoptosis in Myocardial Ischemia/Reperfusion Injury via Modulating the MiR-145-5p/ROCK1 Axis." Your study addresses an important and relevant area of research with potential implications for novel therapeutic strategies targeting myocardial ischemia/reperfusion injury. The exploration of OIP5-AS1's interaction with miR-145-5p and its downstream impact on ROCK1 is promising and could significantly contribute to the existing literature.

However, the methodology section needs substantial improvement. The descriptions provided are too simplified and lack critical details necessary for replication. Specifically, information regarding the exact concentrations of siRNAs, miRNA mimics, and inhibitors utilized in the experiments is missing. Additionally, you should justify the choice of Lipofectamine 2000 over the more RNA-specific reagent Lipofectamine RNAiMAX. Clarification about whether optimization experiments using "killer" siRNA were conducted would also strengthen your methodological transparency. Including control experiments with Lipofectamine alone would clarify if observed effects are due specifically to the treatments or influenced by reagent toxicity. Also, crucial details such as the quantity of RNA used in qPCR reactions, conditions for replicates (technical or biological), the acceptable Ct variation between replicates, and explicitly mentioning the normalization gene used (presumably snU6) must be clearly stated.

In your results section, findings are presented clearly and accompanied by suitable statistical analyses. Nevertheless, minor typographical errors were identified—for instance, "Conrtol" should be corrected to "Control" in Figure 2. Moreover, please clarify whether control groups containing only Lipofectamine (without siRNAs or mimics) were included to exclude nonspecific reagent effects.

Your discussion section is somewhat superficial and would greatly benefit from a more thorough examination of the study's limitations. It is essential to explicitly address limitations, such as small sample sizes, potential off-target effects of transfection agents, and methodological constraints inherent to the model systems used. Additionally, clearly stating future research directions would significantly strengthen this section. For example, suggesting studies that could confirm your findings in larger or more clinically relevant contexts would substantially enhance the manuscript’s scientific and clinical significance.

Furthermore, you should provide clear steps or recommendations to explore the clinical applicability of your findings. For instance, how might the OIP5-AS1 and miR-145-5p/ROCK1 axis modulation translate into clinical interventions or therapeutic approaches?

Lastly, a minor typographical and grammatical revision is necessary throughout the manuscript to maintain professional standards, such as correcting the spelling of "Conrtol" to "Control" in Figure 2. A professional language review is recommended prior to resubmission.

In conclusion, the manuscript presents valuable and promising insights but requires substantial methodological clarifications, deeper acknowledgment of limitations, and a more comprehensive discussion before consideration for acceptance.

Reviewer #3: The authors showed that a long non-coding RNA OIP5-AS1 is upregulated upon ischemia/reperfusion (I/R)-induced apoptosis, and can act like a sponge for miR-145-5p and release the miR-145-5p target (ROCK1) expression. Both in vitro (culture cells) and in vivo models were used.

I could read until Figure 2 end, but started to fell unconfortable for reading the result section afterwards. This is because Figures are not cited correctly in the Result section of maintext and the writing is poor especially in the latter half.

[Methods section]

“miR-145 mimics and inhibitors” information was not shown.

[Result section]

In the paragraph “MiR-145-5p directly targeted by OIP5-AS1 and expressed at low levels in I/R”, Figure 3B is first cited, 3A cannot be found. Neither to 3C 3D 3F.

“Furthermore, silencing of OIP5-AS1 in I/R significantly upregulated miR-145-5p

expression (Fig. 2E).” – Figure 2E is not showing this evidence.

“These results indicated that miR-145-5p was downregulated in

I/R and was a target of OIP5-AS1.” – Better rephrase. Generally, a transcript is a target of miRNA.

“Western blot analysis

and RT-qPCR were performed to investigate ROCK1 expression. Results showed significant upregulation of ROCK1 in I/R cell models compared with controls (Fig.1F).” – Fig.1F is RNA data. No WB data was shown.

“The luciferase report assay demonstrated that miR-145-5p mimics substantially attenuated the luciferase activity of ROCK1-WT, while ROCK1-MUT showed no changes in response to miR-145-5p elevation (Fig. 3F).” – Need rephrase. More correctly, "no change after addition of miR-145-5p mimics.” It is very confusing that this figure is placed as Fig 3F.

“ROCK1 expression was reduced in I/R by OIP5-AS1 deficiency while miR-145-5p was highly

expressed (Fig. 2F and 2G).” – These data are showing ROCK1 RNA and protein levels. but not miR-145-5p level.

“Altogether, OIP5-AS1 behaved as a ceRNA against miR-145-5p, resulting in ROCK1 upregulation.” – Cannot give such strong conclusion from these data. Just “suggest”.

“ceRNA” appears here for the first time without explanation. What is it.

“Figure 5” appears in the last paragraph, without any explanation of individual panels (5A~E). Very poor and unfriendly writing.

[Discussion section]

“OIP5-AS1. OIP5-AS1 was found to inhibit the protective function of miR-145-5p, and it was confirmed to target miR-145-5p.” – “confirmed” sounds too strong compared to the presented data.

The authors can try OIP5-AS1 overexpression experiment in the absence of IR treatment in vitro and in vivo. “a transcript is targeting a miRNA” is confusing, and better be rephrased, like “ OIP-AS1 can sequester miR-145-5p” or “OIP-AS1 has the target site of miR-145-5p”

“This further supports the relationship between OIP5-AS1and miR-145-5p.” – should explain more about the “relationship” in this sentence.

6. PLOS authors have the option to publish the peer review history of their article (what does this mean? ). If published, this will include your full peer review and any attached files.

**Do you want your identity to be public for this peer review?** For information about this choice, including consent withdrawal, please see our Privacy Policy .

Reviewer #1: No

Reviewer #2: No

Reviewer #3: No

---

## [Author Response · Author response to Decision Letter 1]

3 Apr 2025

Reviewer #1: 

Yang and his colleagues observed that the expression of the long non-coding RNA (lncRNA) OIP5-AS1 and apoptosis were increased in myocardial ischemia/reperfusion (I/R) injury. Based on their previous study, they further confirmed that the expression of miR-145-5p was increased, while its target, ROCK1, was decreased. They then demonstrated that reducing OIP5-AS1 expression led to a decrease in apoptosis and experimentally validated the relationship between OIP5-AS1 and miR-145-5p.

The paper presents a very straightforward mechanism, and the experimental evidence supporting each claim is convincing.

However, to strengthen the authors' argument, I suggest including additional experiments to clarify the relationship between ROCK1 and OIP5-AS1:

1. Does overexpression of ROCK1 in the context of reduced OIP5-AS1 modulate the level of apoptosis?

We sincerely appreciate the reviewer’s insightful suggestion to further strengthen the mechanistic link between OIP5-AS1 and ROCK1 in regulating apoptosis. We overexpressed ROCK1 in OIP5-AS1-deficient cells. In Fig7, while OIP5-AS1 knockdown alone reduced apoptosis, concomitant ROCK1 overexpression reversed this effect.

2. How does apoptosis change when anti-miR-145-5p is introduced in a setting where OIP5-AS1 is downregulated?

In Fig6 of the manuscript, we investigated whether OIP5-AS1 regulates ischemia/reperfusion (I/R)-induced apoptosis through miR-145-5p. TUNEL staining results demonstrated that suppression of OIP5-AS1 significantly reduced cellular apoptosis, whereas treatment with a miR-145-5p inhibitor completely abolished this anti-apoptotic effect.

Reviewer #2: 

Thank you for the opportunity to review your manuscript titled "Downregulation of OIP5-AS1 Inhibits Apoptosis in Myocardial Ischemia/Reperfusion Injury via Modulating the MiR-145-5p/ROCK1 Axis." Your study addresses an important and relevant area of research with potential implications for novel therapeutic strategies targeting myocardial ischemia/reperfusion injury. The exploration of OIP5-AS1's interaction with miR-145-5p and its downstream impact on ROCK1 is promising and could significantly contribute to the existing literature.

However, the methodology section needs substantial improvement. The descriptions provided are too simplified and lack critical details necessary for replication. Specifically, information regarding the exact concentrations of siRNAs, miRNA mimics, and inhibitors utilized in the experiments is missing. Additionally, you should justify the choice of Lipofectamine 2000 over the more RNA-specific reagent Lipofectamine RNAiMAX. Clarification about whether optimization experiments using "killer" siRNA were conducted would also strengthen your methodological transparency. Including control experiments with Lipofectamine alone would clarify if observed effects are due specifically to the treatments or influenced by reagent toxicity. Also, crucial details such as the quantity of RNA used in qPCR reactions, conditions for replicates (technical or biological), the acceptable Ct variation between replicates, and explicitly mentioning the normalization gene used (presumably snU6) must be clearly stated.

Thank you for your valuable advice. We acknowledge the need for a more detailed and transparent methodology section to ensure reproducibility and clarity.

We have added specific information regarding the use of siRNAs, miRNA mimics, and inhibitors in the manuscript. Lipofectamine 2000 was chosen for its proven efficacy, compatibility with our cell model, and alignment with prior research methodologies. This decision was further supported by optimization experiments and appropriate controls to ensure the reliability of our findings.

Optimization studies were performed using a cytotoxicity-inducing siRNA (a non-targeting siRNA with a known cytotoxic effect) to establish the optimal transfection conditions while minimizing cell toxicity. The final transfection conditions were selected based on achieving a knockdown efficiency exceeding 80% while maintaining cell viability above 90%.

For qPCR experiments, Acceptable Ct variation between replicates: ≤0.5. U6 was used as the endogenous control for normalization of miRNA expression levels. For mRNA quantification, Actin was used as the housekeeping gene.

In your results section, findings are presented clearly and accompanied by suitable statistical analyses. Nevertheless, minor typographical errors were identified—for instance, "Conrtol" should be corrected to "Control" in Figure 2. Moreover, please clarify whether control groups containing only Lipofectamine (without siRNAs or mimics) were included to exclude nonspecific reagent effects.

Regarding the typographical error in Figure , we acknowledge the mistake and confirm that "Conrtol" will be corrected to "Control" in the revised version of the manuscript. We will thoroughly proofread the entire document to ensure no similar errors remain. Thank you again for your attention to detail and constructive suggestions. And we confirm that control groups containing only Lipofectamine (without siRNAs or mimics) were indeed included in all experiments.

Your discussion section is somewhat superficial and would greatly benefit from a more thorough examination of the study's limitations. It is essential to explicitly address limitations, such as small sample sizes, potential off-target effects of transfection agents, and methodological constraints inherent to the model systems used. Additionally, clearly stating future research directions would significantly strengthen this section. For example, suggesting studies that could confirm your findings in larger or more clinically relevant contexts would substantially enhance the manuscript’s scientific and clinical significance.

Furthermore, you should provide clear steps or recommendations to explore the clinical applicability of your findings. For instance, how might the OIP5-AS1 and miR-145-5p/ROCK1 axis modulation translate into clinical interventions or therapeutic approaches?

We sincerely appreciate the reviewer’s constructive feedback. In response to the concerns raised, we have significantly expanded the limitations and future directions sections in the discussion section of the revised manuscript to address these critical points in greater depth.

Lastly, a minor typographical and grammatical revision is necessary throughout the manuscript to maintain professional standards, such as correcting the spelling of "Conrtol" to "Control" in Figure 2. A professional language review is recommended prior to resubmission.

Thank you for your careful review and valuable feedback. We appreciate your positive comments on the clarity of our results and the statistical analyses. Regarding the typographical error in Figure , we acknowledge the mistake and confirm that "Conrtol" will be corrected to "Control" in the revised version of the manuscript. We will thoroughly proofread the entire document to ensure no similar errors remain. Thank you again for your attention to detail and constructive suggestions.

In conclusion, the manuscript presents valuable and promising insights but requires substantial methodological clarifications, deeper acknowledgment of limitations, and a more comprehensive discussion before consideration for acceptance.

We sincerely appreciate the reviewer’s constructive feedback. In response to the concerns raised, we have significantly expanded the limitations and future directions sections in the discussion section of the revised manuscrip.

Reviewer #3: 

The authors showed that a long non-coding RNA OIP5-AS1 is upregulated upon ischemia/reperfusion (I/R)-induced apoptosis, and can act like a sponge for miR-145-5p and release the miR-145-5p target (ROCK1) expression. Both in vitro (culture cells) and in vivo models were used.

I could read until Figure 2 end, but started to fell unconfortable for reading the result section afterwards. This is because Figures are not cited correctly in the Result section of maintext and the writing is poor especially in the latter half.

[Methods section]

“miR-145 mimics and inhibitors” information was not shown.

Thank you for pointing this out. We apologize for the oversight. The specific details regarding the miR-145 mimics and inhibitors have now been added to the revised manuscript.

[Result section]

In the paragraph “MiR-145-5p directly targeted by OIP5-AS1 and expressed at low levels in I/R”, Figure 3B is first cited, 3A cannot be found. Neither to 3C 3D 3F.

“Furthermore, silencing of OIP5-AS1 in I/R significantly upregulated miR-145-5p

expression (Fig. 2E).” – Figure 2E is not showing this evidence.

“These results indicated that miR-145-5p was downregulated in

I/R and was a target of OIP5-AS1.” – Better rephrase. Generally, a transcript is a target of miRNA.

“Western blot analysis

and RT-qPCR were performed to investigate ROCK1 expression. Results showed significant upregulation of ROCK1 in I/R cell models compared with controls (Fig.1F).” – Fig.1F is RNA data. No WB data was shown.

We sincerely thank the reviewer for pointing out the importance of ensuring consistency between figure citations and the main text descriptions. We have carefully reviewed all figure citations throughout the manuscript and we have verified that all figures are cited in the correct order and context within the text.We have revised the text to ensure that the descriptions of each figure panel are accurate and complete.

“The luciferase report assay demonstrated that miR-145-5p mimics substantially attenuated the luciferase activity of ROCK1-WT, while ROCK1-MUT showed no changes in response to miR-145-5p elevation (Fig. 3F).” – Need rephrase. More correctly, "no change after addition of miR-145-5p mimics.” It is very confusing that this figure is placed as Fig 3F.

The original sentence has been revised to: “The luciferase report assay demonstrated that miR-145-5p mimics significantly reduced the luciferase activity of ROCK1-WT, whereas no change was observed in ROCK1-MUT after the addition of miR-145-5p mimics.”

“ROCK1 expression was reduced in I/R by OIP5-AS1 deficiency while miR-145-5p was highly

expressed (Fig. 2F and 2G).” – These data are showing ROCK1 RNA and protein levels. but not miR-145-5p level.

“Altogether, OIP5-AS1 behaved as a ceRNA against miR-145-5p, resulting in ROCK1 upregulation.” – Cannot give such strong conclusion from these data. Just “suggest”.

Thank you for your careful reading and constructive feedback. We agree that the data presentation and conclusions need to be more precise.We have further verified and confirmed that all figure/table citations are consistent with their descriptions in the main text. And we carefully revised the statement in question to better reflect the preliminary nature of our findings: “Collectively, these data suggest OIP5-AS1 may behave as a ceRNA against miR-145-5p, resulting in ROCK1 upregulation.”

“ceRNA” appears here for the first time without explanation. What is it.

“Figure 5” appears in the last paragraph, without any explanation of individual panels (5A~E). Very poor and unfriendly writing.

We sincerely appreciate the reviewer's valuable feedback regarding the clarity of our manuscript. We have added a clear definition when first introducing the term:“ceRNA” . We have completely rewritten the paragraph to provide a comprehensive explanation of Figure 9.

[Discussion section]

“OIP5-AS1. OIP5-AS1 was found to inhibit the protective function of miR-145-5p, and it was confirmed to target miR-145-5p.” – “confirmed” sounds too strong compared to the presented data.

We sincerely appreciate the reviewer's valuable feedback regarding the clarity of our manuscript. We have rewritten the discussion section. For example, we have carefully revised the statement in question to better reflect the preliminary nature of our findings: Our findings indicate that OIP5-AS1 negatively regulates the cardioprotective effects of miR-145-5p through direct interaction, as demonstrated by our mechanistic studies.

The authors can try OIP5-AS1 overexpression experiment in the absence of IR treatment in vitro and in vivo. “a transcript is targeting a miRNA” is confusing, and better be rephrased, like “ OIP-AS1 can sequester miR-145-5p” or “OIP-AS1 has the target site of miR-145-5p”

Thank you for your careful reading and constructive feedback. We have rewritten the discussion section.

“This further supports the relationship between OIP5-AS1and miR-145-5p.” – should explain more about the “relationship” in this sentence.

Thank you for your careful reading and constructive feedback. We have rewritten the discussion section. We have carefully revised the statement: This finding further supports the competitive binding relationship between OIP5-AS1 and miR-145-5p, where OIP5-AS1 acts as a molecular sponge to miR-145-5p.

---

## [Decision Letter · Decision Letter 1]

30 Apr 2025

PONE-D-25-10451R1Downregulation of  OIP5-AS1 Inhibits Apoptosis in Myocardial Ischemia/Reperfusion Injury via Modulating the MiR-145-5p/ROCK1 AxisPLOS ONE

Dear Dr. Zhang,

Thank you for submitting your manuscript to PLOS ONE. After careful consideration, we feel that it has merit but does not fully meet PLOS ONE’s publication criteria as it currently stands. Therefore, we invite you to submit a revised version of the manuscript that addresses the points raised during the review process.

 Please verify minor language and typo corrections suggested for reviewers, and run a complete proofreading of the manuscript.

We look forward to receiving your revised manuscript.

Kind regards,

Alexis G. Murillo Carrasco

Academic Editor

PLOS ONE

Journal Requirements:

Reviewers' comments:

Reviewer's Responses to Questions

**Comments to the Author**

1. If the authors have adequately addressed your comments raised in a previous round of review and you feel that this manuscript is now acceptable for publication, you may indicate that here to bypass the “Comments to the Author” section, enter your conflict of interest statement in the “Confidential to Editor” section, and submit your "Accept" recommendation.

Reviewer #1: All comments have been addressed

Reviewer #2: All comments have been addressed

Reviewer #3: (No Response)

2. Is the manuscript technically sound, and do the data support the conclusions?

Reviewer #1: Yes

Reviewer #2: Yes

Reviewer #3: Yes

3. Has the statistical analysis been performed appropriately and rigorously? 

Reviewer #1: Yes

Reviewer #2: Yes

Reviewer #3: Yes

4. Have the authors made all data underlying the findings in their manuscript fully available?

Reviewer #1: Yes

Reviewer #2: Yes

Reviewer #3: Yes

5. Is the manuscript presented in an intelligible fashion and written in standard English?

Reviewer #1: Yes

Reviewer #2: Yes

Reviewer #3: Yes

6. Review Comments to the Author

Reviewer #1: Thank you for conducting all the experiments I suggested and for including the corresponding results.

Reviewer #2: I have carefully evaluated the revised manuscript entitled "Downregulation of OIP5-AS1 Inhibits Apoptosis in Myocardial Ischemia/Reperfusion Injury via Modulating the MiR-145-5p/ROCK1 Axis."

I appreciate the authors' efforts to thoroughly address all the concerns raised in the initial review. The manuscript has undergone significant improvements, particularly in the following aspects:

Methodology: The authors have now provided detailed descriptions of the experimental procedures, including the concentrations of siRNAs, mimics, and inhibitors, the rationale for choosing Lipofectamine 2000, and optimization details. The methodology is now sufficiently transparent to allow reproducibility.

Controls: Appropriate negative controls were included and clearly described, addressing concerns about potential nonspecific effects of transfection reagents.

Results Presentation: Minor typographical errors, such as the correction of "Conrtol" to "Control," have been rectified.

Discussion and Limitations: The authors have expanded the discussion significantly, explicitly acknowledging key limitations and proposing valuable future research directions, which strengthen the manuscript's scientific rigor and translational relevance.

Language and Style: The manuscript's language has been improved considerably. Although minor editorial polishing could further enhance clarity, it does not hinder comprehension or scientific value.

Given the authors' careful and thoughtful revisions, I find that all my initial concerns have been satisfactorily addressed.

Therefore, I recommend the manuscript for acceptance after minor language editing (optional proofreading).

Congratulations to the authors for their excellent work and for the significant improvements made to the manuscript.

Reviewer #3: Many points were addressed by the authors.

I just give some points that would need correction before publication.

Table 1

si-OIP5-AS1 and miR mimic‑NC contain "T" in their sequence. If true, why control does not have them.

"Concentration" cannot be understood. Final concentration in culture?

Table 2

Oligos for miRNA would not be correct. Probably the reverse one is an adaptor for RT. Universal Rv primer should be used here.

I have no idea if "miR, microRNA; ROCK1, Rho-associated coiled-coil-containing kinase 1" included in Table 2 is needed in this position.

Line 246

Sentence stopped in the middle. "compared" to what.

Line 268

I didn't get what "pivoral" indicates here.

Line 275

Figure 5 and "8"

8A and 8B are not cited, although authors stated all Figures are to be correctly cited.

Figure legends for Fig1~9 were provided as supporting information.

7. PLOS authors have the option to publish the peer review history of their article (what does this mean? ). If published, this will include your full peer review and any attached files.

**Do you want your identity to be public for this peer review?** For information about this choice, including consent withdrawal, please see our Privacy Policy .

Reviewer #1: No

Reviewer #2: No

Reviewer #3: No

---

## [Author Response · Author response to Decision Letter 2]

1 May 2025

Reviewer #1: Thank you for conducting all the experiments I suggested and for including the corresponding results.

We sincerely appreciate your insightful comments and are grateful for your acknowledgment of our work in implementing all suggested experiments. Your expertise has significantly strengthened our study.

Reviewer #2: I have carefully evaluated the revised manuscript entitled "Downregulation of OIP5-AS1 Inhibits Apoptosis in Myocardial Ischemia/Reperfusion Injury via Modulating the MiR-145-5p/ROCK1 Axis."

I appreciate the authors' efforts to thoroughly address all the concerns raised in the initial review. The manuscript has undergone significant improvements, particularly in the following aspects:

Methodology: The authors have now provided detailed descriptions of the experimental procedures, including the concentrations of siRNAs, mimics, and inhibitors, the rationale for choosing Lipofectamine 2000, and optimization details. The methodology is now sufficiently transparent to allow reproducibility.

Controls: Appropriate negative controls were included and clearly described, addressing concerns about potential nonspecific effects of transfection reagents.

Results Presentation: Minor typographical errors, such as the correction of "Conrtol" to "Control," have been rectified.

Discussion and Limitations: The authors have expanded the discussion significantly, explicitly acknowledging key limitations and proposing valuable future research directions, which strengthen the manuscript's scientific rigor and translational relevance.

Language and Style: The manuscript's language has been improved considerably. Although minor editorial polishing could further enhance clarity, it does not hinder comprehension or scientific value.

Given the authors' careful and thoughtful revisions, I find that all my initial concerns have been satisfactorily addressed.

Therefore, I recommend the manuscript for acceptance after minor language editing (optional proofreading).

Congratulations to the authors for their excellent work and for the significant improvements made to the manuscript.

We are deeply grateful for your insightful comments that significantly improved our manuscript. We have implemented the minor language edits as suggested.

Reviewer #3: Many points were addressed by the authors.

I just give some points that would need correction before publication.

Table 1

si-OIP5-AS1 and miR mimic‑NC contain "T" in their sequence. If true, why control does not have them.

"Concentration" cannot be understood. Final concentration in culture?

We sincerely appreciate the reviewer's meticulous review. SiRNA sequences are shown as DNA templates (containing Thymine). MiRNA mimics/inhibitors are direct synthetic RNA oligos (containing Uracil). We have modified the si-NC sequence to be consistent with DNA template notation (using Thymine instead of Uracil).

We apologize for the ambiguity. The concentrations listed are indeed the final working concentrations in cell culture medium after transfection complex delivery. We have revised the table header to ‘Final working concentration (nM)’.

Table 2

Oligos for miRNA would not be correct. Probably the reverse one is an adaptor for RT. Universal Rv primer should be used here.

I have no idea if "miR, microRNA; ROCK1, Rho-associated coiled-coil-containing kinase 1" included in Table 2 is needed in this position.

We sincerely appreciate the reviewer's expert insights regarding Table 2. We have removed the redundant reverse primer sequence and moved "miR, microRNA; ROCK1..." to the general abbreviation list in the Table 2 footer.

Line 246

Sentence stopped in the middle. "compared" to what.

We sincerely appreciate the reviewer's careful reading. Regarding the incomplete sentence at Line 246 in the original manuscript, we have revised the full sentence “Compared with the control group, expression of miR-145-5p in H9c2 cardiomyocytes was significantly downregulated in I/R.”

Line 268

I didn't get what "pivoral" indicates here.

We sincerely appreciate the reviewer's careful reading and constructive feedback. We have modified the text to “The results suggested that ROCK1 played a significant role in mediating OIP5-AS1's effects on cardiomyocyte apoptosis in I/R.”

Line 275

Figure 5 and "8"

8A and 8B are not cited, although authors stated all Figures are to be correctly cited.

We sincerely appreciate the reviewer's careful reading and constructive suggestions. We have now ensured all figure panels are properly cited throughout the text. The relevant paragraph now reads “ In the present study, we initially corroborated the efficacy of OIP5-AS1 knockdown in our experimental models (Fig. 5A and 8A). Importantly, OIP5-AS1 deficiency led to a substantial decrease in ROCK1 expression, accompanied by an increase in miR-145-5p levels under I/R conditions (Fig. 5B-C and 8B-C). ”(Line 282)

Figure legends for Fig1~9 were provided as supporting information.

We sincerely appreciate the reviewer's careful reading. We have now uploaded all figure legends (for Fig1-9) as Supporting Information S4 File.

---

## [Editor Report · Decision Letter 2]

4 May 2025

Downregulation of  OIP5-AS1 Inhibits Apoptosis in Myocardial Ischemia/Reperfusion Injury via Modulating the MiR-145-5p/ROCK1 Axis

PONE-D-25-10451R2

Dear Dr. Zhang,

We’re pleased to inform you that your manuscript has been judged scientifically suitable for publication and will be formally accepted for publication once it meets all outstanding technical requirements.

Kind regards,

Alexis G. Murillo Carrasco

Academic Editor

PLOS ONE
---

## [Editor Report · Acceptance letter]

PONE-D-25-10451R2

PLOS ONE

Dear Dr. Zhang,

I'm pleased to inform you that your manuscript has been deemed suitable for publication in PLOS ONE. Congratulations! Your manuscript is now being handed over to our production team.

Kind regards,

on behalf of

Dr. Alexis G. Murillo Carrasco

Academic Editor

PLOS ONE